

# Seasonal variations of the ichthyoplankton assemblage in the Yangtze Estuary and its relationship with environmental factors

Hui Zhang[1,2,3], Weiwei Xian[1,2,3] and Shude Liu[1]

[1] CAS Key Laboratory of Marine Ecology and Environmental Sciences, Institute of Oceanology, Chinese Academy of Sciences, Qingdao, China
[2] Laboratory for Marine Ecology and Environmental Science, Qingdao National Laboratory for Marine Science and Technology, Qingdao, China
[3] Center for Ocean Mega-Science, Chinese Academy of Sciences, Qingdao, China

Corresponding authors
Hui Zhang, zhanghui@qdio.ac.cn
Weiwei Xian, wwxian@qdio.ac.cn

## ABSTRACT

Seasonal variations of the ichthyoplankton assemblage and its relationship with the environment were analyzed based on four seasonal surveys during 2012. Historical data was collected to be compared with results from previous years in order to indicate the seasonal and inter-annual variation of the ichthyoplankton assemblage in the Yangtze Estuary and the adjacent waters. A total of 3,688 individuals belonging to 5 orders, 9 families, and 15 species were collected. No samples were collected in the winter cruise. In 2012, all samples were separated into four ecotypes, which is comparable with the historical data from previous years. The *Engraulis japonicus* was the most abundant species of all teleost fishes. The *E. japonicus* was captured in every season and contributed the most to the abundance of ichthyoplankton, which was greater than that of previous years. This result may be due to the periodic fluctuations of *E. japonicus* or from the displacement of spawning grounds offshore for environmental reasons. The diversity indices of the assemblage were significantly different among seasons, with the number and abundance of the species peaking in the spring, while richness, evenness and diversity indices peaked in the autumn. The species richness of the ichthyoplankton varied from 0.74 to 1.62, the Pielou evenness index varied from 0.10 to 0.49 and the Shannon–Wiener index varied from 0.19 to 1.04. The results of CCA analysis showed that the major factors affecting the ichthyoplankton assemblage differed throughout the seasons. Chla was the key factor affecting the ichthyoplankton in 2012. These seasonal and inter-annual variations likely resulted from migrations associated with fish spawning as well as the environment. Compared with data from previous studies, the relationship between the assemblage structure of ichthyoplankton and corresponding environmental variables have undergone a decline.

# INTRODUCTION

The Yangtze Estuary is a transitional region between freshwater and the sea, which is in an advantageous geographical location with a distinct ecological environment. Profiting from the freshwater runoff of the Yangtze River, the Taiwan Warm Current, the East China

Sea Coastal Current, and the Yellow Sea Coast Current, the Yangtze River has become an excellent spawning and nursing ground for an array of economic fish species and a crucial fishery ground in China (*Luo & Shen, 1994*). However, the Yangtze River basin, especially the estuary area, is characterized by a high level of industrialization and urbanization (*Chai et al., 2009*), exposing the estuary to anthropogenic agents from the populated areas and industries upstream of the estuary. The construction and operation of the Three Gorges Reservoir has resulted in short-term and long-term impacts not only on the ecosystem of the Yangtze Estuary, but also in the distribution and community structure of marine organisms (*Xian, Liu & Luo, 2004*). Due to the intensity of trawling operations and environmental pollution, the structure of marine fishery resources was characterized by a recession in economic fish species as well as the variety of species (*Shan & Jin, 2011*). Therefore, it is necessary to determine the relationship between the ichthyoplankton assemblage and the environmental variation.

The spatial and temporal variation of ichthyoplankton assemblages has been widely studied in the field of marine ecology (such as *Yang, Wu & Sun, 1990*; *Zhu, Liu & Sha, 2002*; *Zhong, Wu & Lian, 2007*; *Zhang, Xian & Liu, 2015*; *Zhang, Xian & Liu, 2016*). With better insight into the state of the Yangtze Estuary, many domestic scholars have studied the seasonal variation in the composition and biodiversity of species, as well as the characteristics of the ichthyoplankton assemblage structure and its relationship with environmental factors such as the depth, dissolved oxygen, temperature, and salinity in this region. In the springs of 1999 and 2001, *Liu, Xian & Liu (2008)* reported the taxonomic identification of a total of 11,540 ichthyoplankton individuals in the Yangtze Estuary, which belonged to 11 orders, 18 families, and 32 species. Salinity, depth, dissolved oxygen, and total suspended particulate matter were the major factors affecting the ichthyoplankton assemblages in the study areas. *Wei et al. (2012)* reported that a total of 93 ichthyoplankton samples were collected at 15 stations in Hangzhou Bay in the summers from 2004 to 2010. As a result, 233 eggs and 29,825 larvae were obtained. The correlation was significant between ichthyoplankton logarithm density and factors of hydrological conditions. The goal of this paper is to show the characteristics of the ichthyoplankton assemblage by gathering data on species composition and biodiversity in the Yangtze Estuary from four cruises that took place in 2012, as well as to reveal the relationship between the spatial–temporal distribution patterns in the ichthyoplankton assemblage and environmental factors. The results could provide a scientific basis for the management and sustainable utilization of fishery resources in the Yangtze Estuary.

# MATERIALS AND METHODS

## Data collection

A total of 40 sampling stations were located at the Yangtze Estuary and its adjacent waters (30°45′–32°00′N, 121°00′–123°20′E) (Fig. 1). Samples were collected using the trawl, guided by the "Specification of Oceanographic Investigation" (GB12763-2007) in February, May, August, and November of 2012. This gear has a horizontal opening of 0.8 m and a vertical opening of 2.8 m (mesh size of 0.5 mm). The trawl was monitored

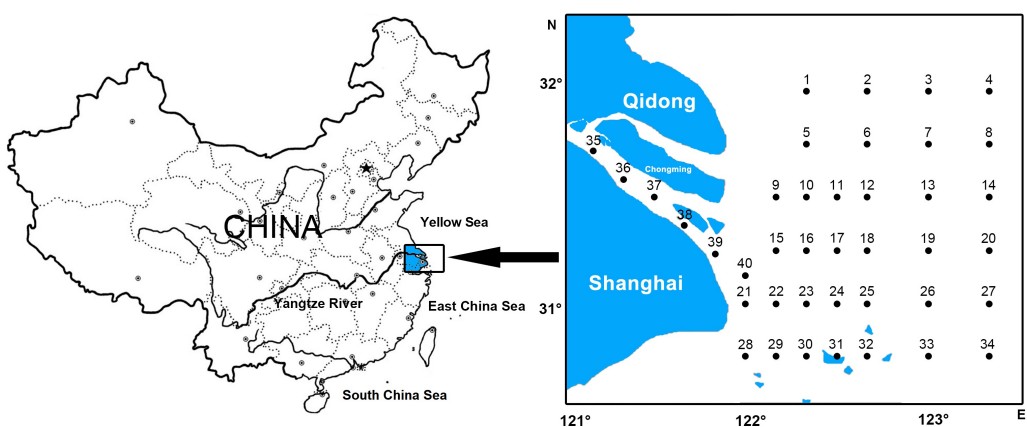

**Figure 1 Location of survey stations of ichthyoplankton in Yangtze estuary.**

horizontally with the vessel speed of approximately 2 knots, lasting 10-min at each station. Samples taken from each trawl were immediately preserved in a 5% formalin buffer for later sorting. Real-time data collected on the environmental parameters of the water column included temperature (T), salinity (S), total nitrogen (TN), total phosphorus (TP), pH, suspended matter (SPM), depth (D), dissolved oxygen (DO), chemical oxygen demand (COD), and chlorophyll a (Chla). The collection of data was under the guidance of the "Specification of Oceanographic Investigation" (GB12763-2007). Field experiments were approved by Three Gorges Project Construction Commission of the State Council, China (Project Numer: JJ2013011).

## Species identification

At the laboratory, fish eggs and larvae were counted and sorted to the lowest possible taxonomic level at each station according to the morphological characteristics found in the literature (*Zhang et al., 1985*; *Cheng & Zheng, 1987*; *Wu, Shao & Lai, 2012*). These were classified into different ecotypes by their distinct ecological habits based on the descriptions in the literature (*Yang, Wu & Sun, 1990*; *Luo & Shen, 1994*). Numerous fish eggs and larvae that lacked clear morphological features could not be identified using this approach, so molecular identification was applied to prevent misidentification.

## Data analyses

The abundance of ichthyoplankton was standardized and expressed as the total number of individual fish eggs and larvae per 10 min-trawling (ind/trawl). The dominant species were determined using the Index of Relative Importance (IRI) developed by *Zhu, Liu & Sha (2002)*:

$$IRI = N * 100\% * F * 100\%.$$

$N * 100\%$ and $F * 100\%$ are the relative abundance and frequency of occurrence, respectively. The IRI of the dominant species should be greater than 100.
The Margalef's richness ($D$), Shannon-Wiener index ($H'$, $\log_e$), and Pielou's evenness ($J'$) were calculated for each station. Related equations were as follows (*Ludwig & Reynolds, 1988*; *Qian & Ma, 1994*):

$$D = (S-1)/\ln N$$

$$H' = -\sum_{i=1}^{S} P_i \cdot \ln P_i$$

$$J' = H'/\ln S.$$

Where "$S$" is the number of species, "$N$" is total individuals, and "$Pi$" is the proportion of fish species individuals to the total individuals.

The homoscedasticity was measured with the method Levene's test, after which a one-way ANOVA was performed to assess the difference in abundance, biomass, species richness, and biodiversity index among four cruises. When a significant difference was detected, the Duncan's test was applied for multiple comparisons. Canonical correspondence analysis (CCA) was applied to analyze the correlation between environmental factors and the distribution pattern of ichthyoplankton assemblages. To eliminate the effects of a few dominant species, numerous zeros in the species data, and a highly variable value in environmental data, all data matrix were transformed by $\log(x+1)$.

All maps were drawn with Surfer 8.0 and statistical analyses were performed with PRIMER 5.0 (PRIMER-E, Plymouth, UK), SPSS 16.0 (SPSS, IBM, Armonk, NY, USA) and CANOCO 4.5 (http://www.canoco5.com/).

# RESULTS

## Species composition

A total of 3688 individuals, including 689 fish eggs and 2,999 larvae from 4 cruises, were sorted. All samples belonged to 7 orders, 12 families, and 15 species including one unidentified species (Table 1). The abundance and biomass of Engraulidae, were dominant in 2012.

According to the habitats and distribution characteristics of ichthyoplankton, 4 ecotypes were included in this study (Table 1):

Fresh water species included *P. engraulis*, which complete their entire life cycle in fresh water. This species is distributed in fresh waters or oligo-salt waters adjacent to the inner sides of the estuary and had the fewest individuals, accounting for 1.97% of the entire abundance in four seasons.

Brackish water species, which use the estuary as a habitat but which complete the early developmental stages in the waters close to the estuary, include catadromous species and anadromous species. These species include *C. nasus*, *C. mystus*, *C. spinosus*, *H. sajori*, and one species belonging to Takifugu, accounting for 11.30% of the total abundance.

Coastal species typically gather in shallow coastal waters for reproduction and development in the spring and summer months and migrate to abyssal regions in winter. Four species were included: *A. commersoni*, *L. polyactis*, *A. bleekeri*, and *M. monodactylus*, accounting for 4.55% of the total abundance.

**Table 1  Presence (+) of species in ichthyoplankton samples in the present study.**

| Species | Code | Ecotype | Month | | | |
|---|---|---|---|---|---|---|
| | | | Feb. | May | Aug. | Nov. |
| **Engraulidae** | | | | | | |
| *Engraulis japonicus* | *Enja* | Marine | | + | + | + |
| *Anchoviella commersoni* | *Anco* | Coastal | | + | + | + |
| *Coilia nasus* | *Cona* | Brackish water | | | + | + |
| *Coilia mystus* | *Comy* | Brackish water | | + | + | + |
| **Cyprinidae** | | | | | | |
| *Pseudolaubuca engraulis* | *Psen* | Fresh water | | | + | |
| **Sciaenidae** | | | | | | |
| *Larimichthys polyactis* | *Lapo* | Coastal | | + | | |
| **Scombridae** | | | | | | |
| *Scomber japonicus* | *Scja* | Marine | | + | | |
| **Trichiuridae** | | | | | | |
| *Trichiurus japonicus* | *Trja* | Marine | | | | + |
| **Atherinidae** | | | | | | |
| *Allanetta bleekeri* | *Albl* | Coastal | | + | + | + |
| **Scorpaenidae** | | | | | | |
| *Minous monodactylus* | *Mimo* | Coastal | | + | + | |
| **Triglidae** | | | | | | |
| *Chelidonichthys spinosus* | *Trfa* | Brackish water | | + | + | + |
| **Hemiramphidae** | | | | | | |
| *Hemiramphus sajori* | *Hesa* | Brackish water | | | | + |
| **Syngnathidae** | | | | | | |
| *Syngnathus acua* | *Syac* | Marine | | | | + |
| **Lophiidae** | | | | | | |
| *Lophius litulon* | *Loli* | Marine | | + | | |
| **Tetraodontidae** | | | | | | |
| *Takifugu* sp. | *Tasp* | Brackish water | | + | | |

The marine species that migrate to the profundal zone (>30 m) for feeding as they hit adulthood then returns to estuary or coastal waters for spawning and breeding includes *E. japonicus, S. japonicus, T. japonicus, S. acua,* and *L. litulon,* which were the greatest contributors to the total abundance, accounting for 82.30%.

The greatest number of species were collected in the spring, including 3 brackish water species, 4 coastal species, and 3 marine species. This was followed by autumn with 9 species collected, including 4 brackish water species, 2 coastal species, and 3 marine species, The least number of species were caught in the summer, including 1 fresh water species, 1 marine species, 3 brackish water species, and 3 coastal species. In the summer, ichthyoplankton assemblages were dominated by brackish water and coastal species, while coastal species and brackish water species were dominant in the spring and autumn months, respectively.

*E. japonicus, C. mystus, A. commersoni, A. bleekeri,* and *C. spinosus* were widespread species, which were captured in all four seasons. Conversely, 7 species (54.55% of the total

**Table 2   Composition of dominant ichthyoplankton species in different seasons.**

| Dominant species | Spring | | Summer | | Autumn | |
|---|---|---|---|---|---|---|
| | IRI | Percentage of quantity (%) | IRI | Percentage of quantity (%) | IRI | Percentage of quantity (%) |
| *Engraulis japonicus* | 5,120.99 | 90.2 | 1,532.11 | 74.61 | 19.74 | 3.95 |
| *Coilia mystus* | 158.99 | 3.6 | 40.37 | 10.01 | 3.29 | 1.32 |
| *Allanetta bleekeri* | 11.27 | 1.13 | 1.15 | 0.46 | 52.63 | 5.26 |
| *Chelidonichthys spinosus* | 4.22 | 0.85 | 1.15 | 0.46 | 3.29 | 1.32 |
| *Anchoviella commersoni* | 2.11 | 0.42 | 1.15 | 0.46 | 26.32 | 5.26 |
| *Larimichthys polyactis* | 1.52 | 0.61 | | | | |
| *Scomber japonicus* | 0.12 | 0.025 | | | | |
| *Minous monodactylus* | 0.12 | 0.025 | | | | |
| *Lophius litulon* | 0.12 | 0.025 | | | | |
| *Takifugu* sp. | 0.12 | 0.025 | | | | |
| *Coilia nasus* | | | 2.30 | 2.02 | 19.74 | 3.95 |
| *Pseudolaubuca engraulis* | | | 300.25 | 12.00 | | |
| *Hemirhamphus sajori* | | | | | 39.47 | 5.26 |
| *Syngnathus acua* | | | | | 13.16 | 2.63 |
| *Trichiurus japonicus* | | | | | 3.29 | 1.32 |

species) were captured only in single season; *S. japonicus, L. polyactis* and *M. monodactylus* were only collected in the spring, *P. engraulis* was only collected in the summer, and *S. acua, H. sajori, T. japonicus* were only captured in the autumn. The distribution pattern of ichthyoplankton assemblages varied with seasonal changes due to the ecological habits of the species as well as their tendencies toward differing seasonal compositions.

The Index of Relative Importance (IRI) was used to discuss the dominant species. Species with an index greater than 1,000 and a range from 100 to 1,000 were considered a dominant species and common species, respectively. These factors combined indicated the important species. A clear variation occurred in the composition of the dominant species in every season (Table 2). *E. japonicus* and *C. nasus* were the dominant species which contributed the most (98.63%) to the total abundance, followed by *E. japonicus* which occupied 93.80% in the spring, as indicated in Table 2. *E. japonicus* occupied the greatest proportion of the abundance in the spring and autumn. The characteristics and composition of the dominant species and the variation in the degree of dominance showed a distinct difference during three investigations, which indicated the seasonal variations in the ichthyoplankton assemblage structure.

## Spatial and temporal variation

The spatial distribution of the ichthyoplankton abundance in the Yangtze Estuary in 2012 showed significant seasonal variation (Fig. 2), with the highest abundance in the spring and the lowest in the autumn.

A total of 2,604 individuals were captured in the spring, including 317 fish eggs and 2,287 larvae. Larvae were widespread, with the exception of the river channel and the northern locations of the investigation areas. The most widely distributed species was

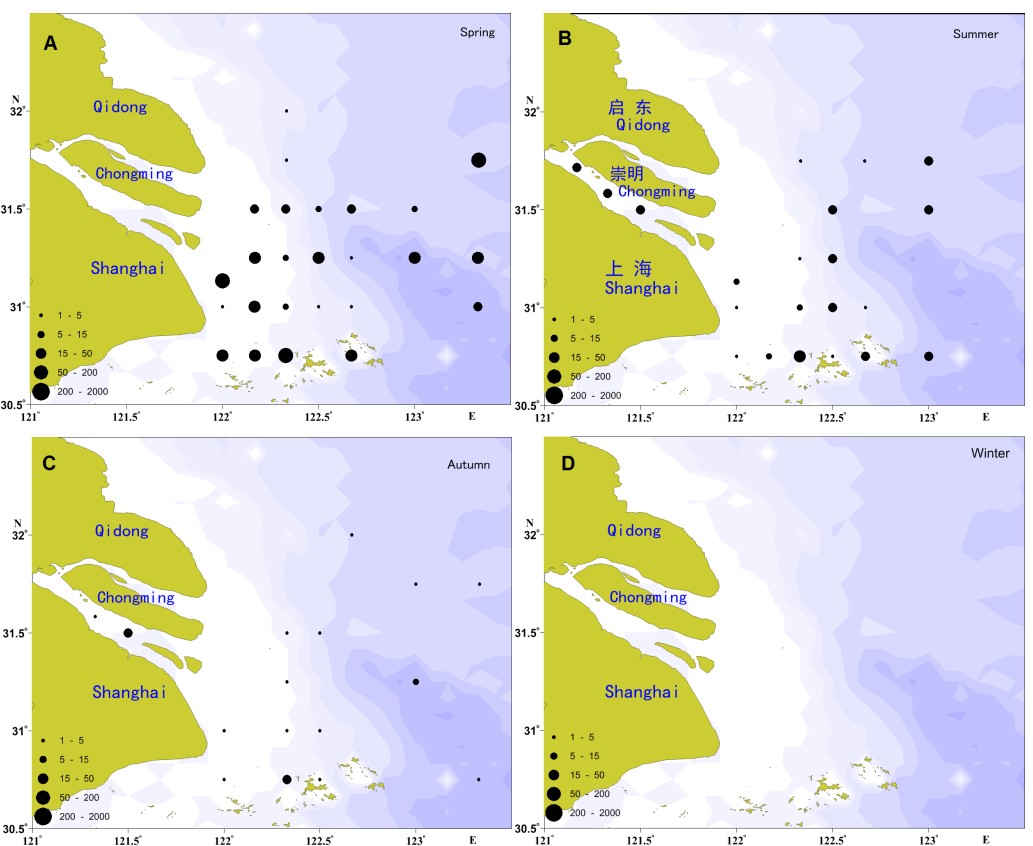

**Figure 2** **Distribution of ichthyoplankton abundance in the present study.** (A) Distribution of ichthyoplankton abundance in spring; (B) distribution of ichthyoplankton abundance in summer; (C) distribution of ichthyoplankton abundance in autumn; (D) distribution of ichthyoplankton abundance in winter.

*E. japonicus*, followed by *A. bleekeri*, *C. mystus*, *C. spinosus*, and *L. polyactis*. In total, 366 individuals were recorded in the summer, including 120 fish eggs and 246 larvae, which were primarily distributed in the river channel and the southern and eastern portions of the study areas. *E. japonicus* had the greatest number of larvae, followed by *P. engraulis* and *C. mystus*, which shared a similar distribution range. Only 76 larvae were collected in the autumn, and no fish eggs; these were mainly distributed in the river channel and the southern portion of the study areas, with the *C. nasus* as the majority, followed by *E. japonicus*, *A. bleekeri*, *A.commersoni*, and *H. sajori*.

## Biodiversity

The statistical result of Levene's test (df1 = 2, df2 = 9, sig = 0.165 > 0.05) indicated that the difference of homoscedasticity of the index is not significant. We then used the One-way ANOVA test for further analysis. The One-way ANOVA test revealed that a significant difference occurred among three diversity indexes during four seasons (df of inter-season = 2, df of intra-season = 9, F = 4.601, P = 0.0095 < 0.01).

According to the results, autumn presented the highest diversity indexes while spring presented with the lowest (Table 3). Furthermore, according to the result of multiple

**Table 3  Diversity index of ichthyoplankton in different seasons.**

|  | Spring | Summer | Autumn | Winter |
|---|---|---|---|---|
| Species richness ($D$) | $0.79 \pm 0.05^A$ | $0.74 \pm 0.08^A$ | $1.62 \pm 0.11^B$ | $0.00 \pm 0.00^C$ |
| Pielou evenness index ($J'$) | $0.10 \pm 0.06^A$ | $0.38 \pm 0.08^B$ | $0.49 \pm 0.05^B$ | $0.00 \pm 0.00^C$ |
| Shannon-Wiener index ($H'$) | $0.19 \pm 0.05^A$ | $0.61 \pm 0.09^B$ | $1.04 \pm 0.10^C$ | $0.00 \pm 0.00^D$ |

**Notes.**

$P < 0.01$, Numbers with different superscript are significantly different with each other.

comparisons, all the diversity indexes had a significant difference between spring and autumn ($P < 0.01$). Furthermore, $H'$ and $J'$ between spring and summer as well as $J'$ between summer and winter also occurred with a significant difference. However, no significant difference was detected for $D$ between spring and summer ($F = 3.24$, $P = 0.10 > 0.05$) as well as for $D$ ($F = 4.30$, $P = 0.08 > 0.05$) and $H'$ ($F = 2.96$, $P = 0.07 > 0.05$) between summer and autumn.

## CCA analysis

The relationships between the environmental factors and the species were clarified in the CCA ordination diagram using the data from 15 species and the set of 10 environmental factors. The first axis (eigenvalues = 0.497) and the second axis (eigenvalues = 0.290) of the CCA plot explained 14.4% of "species data" variation and 65.6% of variation in "species-environment relation". The species-environment correlation coefficients of these two axes were 0.832 and 0.621, respectively. The Monte-Carlo test (Table 4) indicated that Chla was the key environmental factor affecting ichthyoplankton assemblages ($P < 0.05$). As shown in the diagram, the first axis was strongly correlated with Chla, SPM, TP, COD, DO, and the remaining environmental factors displayed a higher correlation with axis 2 than axis 1. SPM, Chla, TP, and temperature exerted a positive effect on the first axis. In addition, depth showed a positive correlation and TN showed a negative correlation with the second axis, respectively. The CCA ordination plot of sampling stations (Fig. 3) revealed that in the spring, stations were located in an area with a significantly higher level of dissolved oxygen, TN, pH, and salinity. The location of sampling stations in the summer were relatively scattered and mainly characterized by higher levels of TN, TP, Chla, SPM, temperature, and dissolved oxygen values. As for the investigation areas in the autumn, sampling stations were mainly distributed in areas with higher pH and salinity values.

As shown in the CCA ordination plot of ichthyoplankton species (Fig. 4), the correlation between environmental factors and the distribution of different species was inconsistent. *E. japonicus* showed a strong relationship with dissolved oxygen and was less affected by the remaining factors, while *S. japonicus* was mainly affected by TP, which indicated that the distribution pattern of different species belonging to the same ecotype may be affected by different environmental factors. *L. polyactis* and *A. commersoni* also revealed a significant positive correlation with dissolved oxygen and were mainly distributed in the region of higher dissolved oxygen content. *C. spinosus* showed a distinct distribution pattern positively associated with the higher value of Chla, SPM, and temperature. The distribution pattern of *C. nasus* was positively correlated with deeper and higher concentration of nutrients areas.

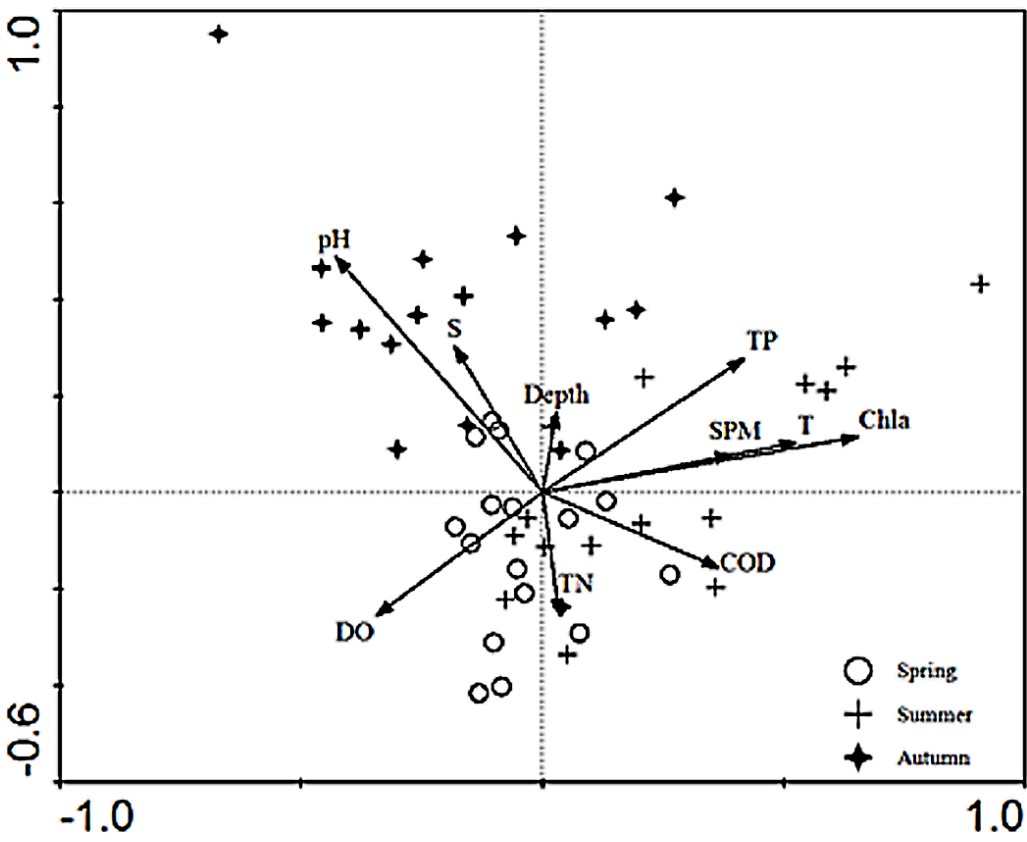

**Figure 3**  CCA biplot of sampling stations.

**Table 4  Conditional effects and correlations of environmental variables with the CCA axes.**

| Environmental factors | Lambda A | P | Axis 1 | Axis 2 |
|---|---|---|---|---|
| Chla | 0.23 | 0.046 | 0.7541 | 0.1330 |
| pH | 0.16 | 0.148 | −0.0108 | 0.0124 |
| DO | 0.13 | 0.334 | −0.0649 | −0.0482 |
| D | 0.1 | 0.414 | 0.0208 | 0.1146 |
| TP | 0.11 | 0.396 | 0.4464 | 0.2961 |
| COD | 0.09 | 0.610 | 0.1636 | −0.0699 |
| SPM | 0.08 | 0.514 | 0.6435 | 0.1287 |
| TN | 0.14 | 0.298 | 0.0251 | −0.1837 |
| T | 0.08 | 0.64 | 0.1195 | 0.0236 |
| S | 0.08 | 0.682 | −0.0912 | 0.1521 |

Species such as *H. sajori* (*hesa*), *A. bleekeri* (*Albl*), *M. monodactylus* (*Mimo*), and *S. acua* (*syca*) had a positive correlation with pH and salinity but *A. bleekeri* had lower demand for pH and salinity than the other three species.

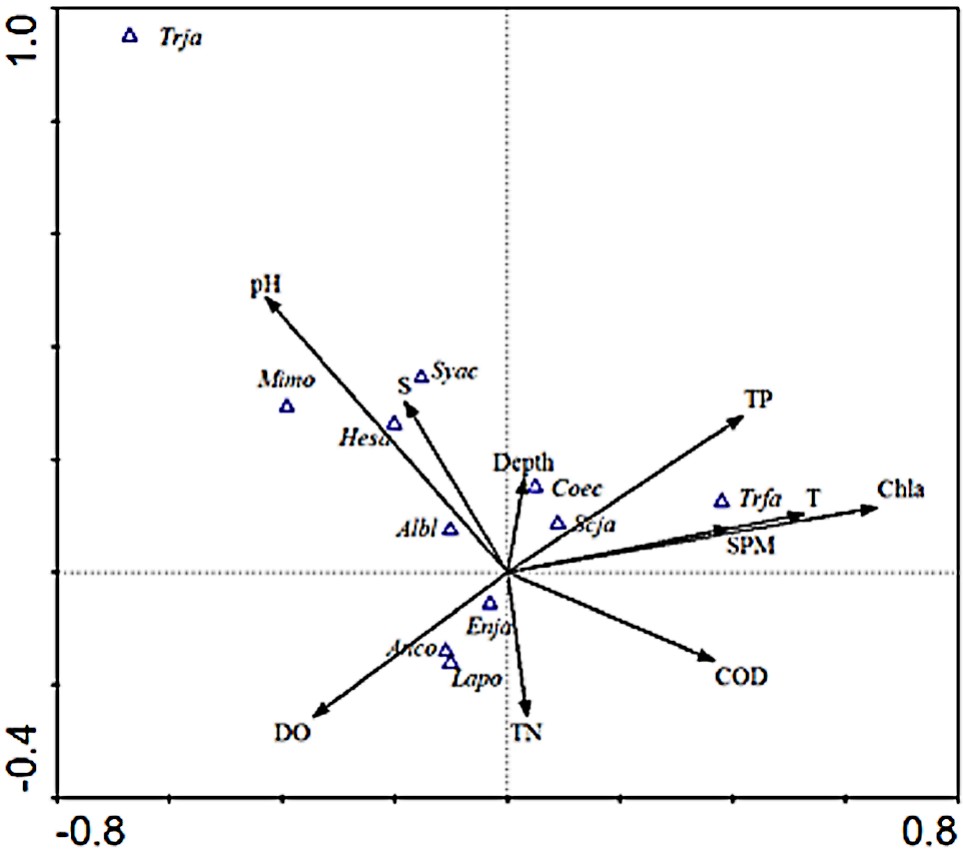

**Figure 4 CCA biplot of ichthyoplankton species.** The italic characters indicate the abbreviate name of the species as shown in Table 1.

## DISCUSSION

The ichthyoplankton assemblages in estuaries are complex both in species composition and distribution. Studies show that the organization of ichthyoplankton in estuarine systems is influenced by the interactive effects of a multitude of biotic and abiotic processes. Biological factors include the location, timing and manner of spawning, larval life history, larval behavior, rates of predation, and feeding (*Leis, 1991*; *Azeiteiro et al., 2006*). Physical factors include salinity (*Whitfield, 1999*), temperature (*Blaxter, 1992*), turbidity (*Islam, Hibino & Tanaka, 2006*), dissolved oxygen (*Rakocinski, LyczkowskiShultz & Richardson, 1996*), depth (*Wantiez, Hamerlin-Vivien & Kulbicki, 1996*), river flow (*Faria, Morais & Chícharo, 2006*), sediment characteristics, and hydrographic events such as currents, winds, eddies, upwelling, and stratification of the water column (*Gray, 1993*). The present study was based on surveys during four seasons in 2012. Our aim was to provide detailed characterizations of the ichthyoplankton assemblage in 2012 and to evaluate the influence of environmental factors on the spatial distribution and intra-annual variations of ichthyoplankton assemblages associated with the Yangtze Estuary.

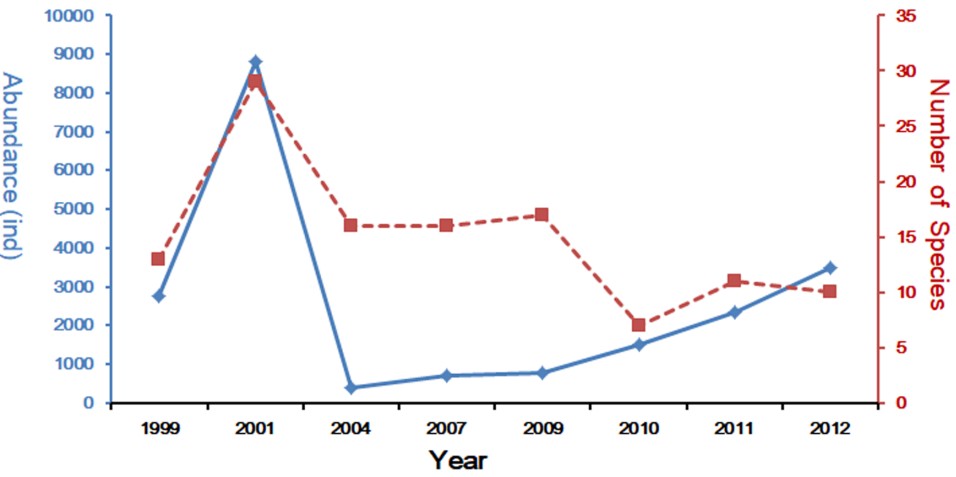

**Figure 5** **Spring long-term variation of the ichthyoplanton in Yangtze Estuary.** The data for 1999–2011 was referenced from *Zhang, Xian & Liu (2015)* and *Zhang, Xian & Liu (2016)*.

## Species composition and seasonal variation

In the last decades, many scholars have reported the community structure and biodiversity of ichthyoplankton assemblages and their relationship with environmental factors. The study of *Yang, Wu & Sun (1990)*, was carried out from 1985 to 1986 with 10 cruises in Yangtze Estuary, collecting 94 species. Another study based on four cruises in 2007 collected 45 species (*Liu & Xian, 2009*), and shared the same investigation area with this study. *Zhang, Xian & Liu (2015)* and *Zhang, Xian & Liu (2016)* studied the ichthyoplankton assemblages in spring (1999–2007) and autumn (1998–2009). In spring, forty-two ichthyoplankton belonging to 23 families were collected. Engraulidae was the most abundant family, including six species and comprising 67.91% of the total catch (*Zhang, Xian & Liu, 2015*), while in autumn a total of 969 ichthyoplankton, constituting 33 species from 19 families and 10 orders, were collected during the seven sampling autumns in the Yangtze Estuary. This sample included 226 fish eggs and 743 larvae and juveniles (*Zhang, Xian & Liu, 2016*). Species composition of ichthyoplankton assemblages in spring showed a descending trend with 20 species in 1999 (*Zhu, Liu & Sha, 2002*), 31 in 2001(*Zhang, Xian & Liu, 2015*), 12 in 2004 (*Zhang, Xian & Liu, 2015*), 17 in 2007 (*Zhang, Xian & Liu, 2015*) and only 10 in 2012 (the present study). The comparison of these studies reveals that species composition of ichthyoplankton assemblages have been suffering a decline in numbers (Fig. 5). Composition and numbers of fresh waters species in this study were less than the investigation results of 2007 (*Liu & Xian, 2009*; *Zhang, Xian & Liu, 2015*); only *P. engrauli* was collected. This phenomenon indicated that the composition and numbers of fresh water species have been experiencing a significant decline due to the impact of anthropogenic activity and a change in the natural environment. *Zhong, Wu & Lian (2007)* has presented that salinity in the Yangtze Estuary and its adjacent waters was significantly influenced by surface runoff, which may result in the decline of fresh water species.

The abundance of ichthyoplankton assemblage in this study was highly concentrated on the dominant species, *E. japonicus,* which was the commonly identified member of the ichthyoplankton assemblages (*Harrison & Whitfield, 1990*; *Whitfield, 1999*). *E. japonicus* resources have declined dramatically based on the comparison in May of 1999, 2001, 2004, and 2007 (*Zhang, Xian & Liu, 2015*), and May and June of 2008 (*Shan & Jin, 2011*). Nevertheless, this study showed that the quantity of *E. japonicus* resources in every season of 2012 was increasing, which was not consistent with the previous results. *Watanabe (2007)* reported that *E. japonicus* resources had relatively steady fluctuations in quantity as a result of climate and environment changes, which were mainly caused by water circulation and fluctuations in temperature. Although Watanabe did not point out the duration of the fluctuation, this conclusion was also supported by the investigation results in this study.

Based on the results of this study and the comparison with the results of other scholars, the community structure of ichthyoplankton assemblages in the Yangtze Estuary have fluctuated greatly over a short period of time .This phenomenon is related to the use of coastal water in different months by dominant species for breeding and feeding (*Shan & Jin, 2011*). However, the sensitivity of the different species to different disruptive factors such as fishing and environmental changes, was also species-dependent and related to ecological niche and habits variations among species.

## Biodiversity and its spatial–temporal variation

In this study, the biodiversity indexes in each season showed significant difference, but all the indexes were relatively low (Table 3). The investigation in 2007 (*Zhang, Xian & Liu, 2015*; *Zhang, Xian & Liu, 2016*) collected 52 fish eggs and 638 larvae in the spring, 3973 fish eggs and 1,342 larvae in the summer, and 6 fish eggs and 450 larvae in the autumn, which were significantly greater than the results in this study in terms of quantity and species composition. Significant differences existed in the spatial and temporal variation of ichthyoplankton assemblages and biodiversity in the Yangtze Estuary from 2007 to 2012. Furthermore, the composition of dominant species in different seasons presented clear variation between 2007 and 2012. In 2007, *A. bleekeri, C. mystus,* and *E. japonicus* were the dominant species in the spring, *E. japonicus, C. mystus,* and *S. elongata* were the dominant species in the summer, and *H. prognathous, E. japonicus* and *C. stigmatias* were the dominant species in the autumn. As for 2012: *E. japonicus* was the dominant species in the spring, *E. japonicus* and *C. nasus* were the dominant species in the summer, and *A. bleekeri* was the dominant species in the autumn.

Most of the marine fish's spawning season occurred in the spring and summer, which brings the abundance of ichthyoplankton assemblages to their maximum point in this period (*Young & Potter, 2003*; *Sabatés et al., 2007*). In this study, the abundance and the number of species were highest in the spring, which was consistent with the above conclusion. However, due to the high concentration of *E. japonicus* in the spring, accounting for 93.80% of the total abundance, the diversity indexes in the spring was less than summer and autumn.

The seasonal variation of biodiversity in the estuary is mainly dependent upon the selection of reproductive areas by grown fish (*Hernández Miranda, Palma & Ojeda, 2003*)

as well as the influence of seasonal variation on the water environment in the spawning area on spawning behavior (*Lam, 1983*). With the increasing intensity of fishing, a variety of fish reached sexual maturity earlier, which led to the spawning period occurring earlier than before. This effect caused the peak abundance value of ichthyoplankton to occur earlier than before, which may be one of the reasons for seasonal variation in ichthyoplankton abundance in the Yangtze Estuary. The environment in the Yangtze Estuary is complicated and changeable with rapid, drastic fluctuations (*Luo & Shen, 1994*), which prevents the fish from adapting to environmental changes in time, hence the assemblage biodiversity of ichthyoplankton was relatively low.

### Relationship between the distribution of ichthyoplankton assemblages and environmental factors

Distribution pattern of ichthyoplankton assemblages in the estuary were affected by both abiotic factors and environmental factors (*Zhu, Liu & Sha, 2002*). In general, salinity was the major factor which determined the structural changes of plankton communities in the estuaries (*Wooldridge, 1999*). Due to the specific geographical conditions and the inflow of fresh water into the estuary, salinity showed a clear gradient corresponding to the direction of the runoff. Ichthyoplankton assemblages altered according to the variation of salinity content. CCA ordination results indicated that the key factor affecting the assemblage structure of ichthyoplankton was not salinity but Chla, results that conflicted with studies from *Kushlan (1976)* and *Thiel et al. (1995)*. The salinity condition in the Yangtze Estuary was significantly influenced by the surface runoff which could result in the decline of fresh water species (*Zhong, Wu & Lian, 2007*). In the present work, only one fresh water species was collected and other species are not sensitive to the variation of salinity. This may be the reason that salinity is not the key factor. Due to the fluctuation of the water environment and species composition in different season, the influence of environmental factors was not consistent among different seasons and years. *Harris, Cyrus & Beckley (1999)* presented that DO was the leading indicator accounting for the variation of the community structure and abundance of ichthyoplankton assemblages, which was also correspond with conclusions made by other scholars (*Castillo-Rivera, Zavala-Hurtado & Zárate, 2002*). In this study, DO was also confirmed as the dominant factor affecting the assemblage structure in the Yangtze Estuary. Temperature, nutrient content, COD, pH also made significant contributions to the assemblage structures. The CCA ordination only explained 14.4% and 65.6% of the variation in species and environment, respectively, thus, more biotic factors and environmental factors need to be collected in later investigations to understand the environmental-biological relationships.

## CONCLUSION

Across the four surveys conducted in 2012, 3,688 individuals of 15 species were collected. We found that significant seasonal differences occurred in the species number and abundance of ichthyoplankton assemblages in the Yangtze Estuary with low biodiversity. Chla was the key environmental factor affecting the assemblage structure of ichthyoplankton in 2012, which was different than the results of previous research.

With the rapid development of industrialization, urbanization, and marine fishery, ichthyoplankton resources are declining significantly, which may strengthen the trend toward the simplification of fishery resources in the Yangtze Estuary. Protection of fishery resources and continuous tracking and monitoring are imperative in the Yangtze Estuary.

## ACKNOWLEDGEMENTS

We thank all the editors and reviewers for their constructive comments on the present work.

### Funding

This work was supported by the National Natural Science Foundation of China (No. 31872568), the International Partnership Program of Chinese Academy of Sciences (CAS-JSPS, GJHZ1885), the Natural Science Foundation of China—Shandong Joint Fund for Marine Ecology and Environmental Sciences (U1606404), and the Youth Talent Support Program of the Laboratory for Marine Ecology and Environmental Science, Pilot National Laboratory for Marine Science and Technology (Qingdao) (LMEES-YTSP-2018-01-12). The funders had no role in study design, data collection and analysis, decision to publish, or preparation of the manuscript.

### Grant Disclosures

The following grant information was disclosed by the authors:
National Natural Science Foundation of China: 31872568.
International Partnership Program of Chinese Academy of Sciences: CAS-JSPS, GJHZ1885.
Natural Science Foundation of China—Shandong Joint Fund for Marine Ecology and Environmental Sciences: U1606404.
The Youth Talent Support Program of the Laboratory for Marine Ecology and Environmental Science, Pilot National Laboratory for Marine Science and Technology (Qingdao): LMEES-YTSP-2018-01-12.

### Competing Interests

The authors declare there are no competing interests.

### Author Contributions

- Hui Zhang conceived and designed the experiments, performed the experiments, analyzed the data, contributed reagents/materials/analysis tools, prepared figures and/or tables, authored or reviewed drafts of the paper, approved the final draft.
- Weiwei Xian conceived and designed the experiments, contributed reagents/materials/-analysis tools, authored or reviewed drafts of the paper, approved the final draft.
- Shude Liu performed the experiments, prepared figures and/or tables.

### Field Study Permissions

The following information was supplied relating to field study approvals (i.e., approving body and any reference numbers):

Field experiments were approved by Three Gorges Project Construction Commission of the State Council, China (Project Numer: JJ2013011).

### Data Availability

All the species and environmental factors used in the present work are included in the Supplemental Materials.

### Supplemental Information

Supplemental information for this article can be found online at http://dx.doi.org/10.7717/peerj.6482#supplemental-information.

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
