# Peer review of "Seasonal variations of the ichthyoplankton assemblage in the Yangtze Estuary and its relationship with environmental factors"

_PeerJ, doi:10.7717/peerj.6482_

## Round 0.1 · original submission · Major Revisions

We have now received the comments from four reviewers. They suggested the study is interesting, but a satisfactory revision is needed. Please carefully follow their comments and resubmit.

Reviewer 1 ·

Basic reporting

This manuscript analyzed seasonal variations of the ichthyoplankton and the relationship between the community and the environmental factors. The results are interesting and material and method were reasonable for the present study. However, the English writing and grammar should be modified by a native English speaker. It would have been interesting to compare in detail previous work and previous studies in detail, to see if there were long term trends and so on. This should be done using clear tables and figures comparing present and past data. The results did not state the exact relationship between ichthyoplankton community and environmental factors.

Experimental design

no comment

Validity of the findings

no comment

Reviewer 2 ·

Basic reporting

The article needs to be edited by a native English speaker.
More literature references should be added to explain why you develop the research and to discuss the different rusults with the prior literature.

Experimental design

No comments.

Validity of the findings

No comments.

Additional comments

The manuscript is well and clearly written and provide comprehensive results. I have provided one major concern and several minor comments. I hope my comments improve this manuscript.
In addition, because I am not a native English speaker, I cannot any comments about English grammar. But I recommend authors ask a native English speaker to confirm English of this manuscript.

Line 23-26, the sentence was not closely connected with the last sentence. Maybe can change to "And historical data was also collected to compared with ...".
Line 34, the authors told about "abundant", but I can't find the data.
Line 38-39, "...affecting the ichthyoplankton in 2012" , affect ichthyoplankto what? it's not complete.
Line 40-42, "Compared with previous ,,, a great change", how is the change, or the sentence is meaningless.
Line 52, Insert "but or however"
Line 59, add a close sentence, like "So it's necessary to ...."
Line 60-61, the sentence is not natural, it's not usually to say "the major study", you can change to "It has been widely reported..."
Line 61, after "marine ecology", please add the references.
Line 65, "however, the recent researches was lacking" you can't say like this, and it's not the necessary reason to research.
Line 67, "characterized"?
Line 68, "analyze the relationship between,,," the purpose of the research is not to analyze the relationship, it's to find the main environmental factors through the relationship analysis. please rewrite the sentence.
Line 79-81, since you have Fig 2, how do you calculate the abundance, please add the formula.
Line 81-84, how do you measure the data? please add the equipment or methods
Line 98, how do you calculate IRI, can't find the formula.
Line 117, the structure of the results is too complicated, please simplify and be clear. maybe 3 or 4 parts as composition, diversity, spatial and temporal variation and CCA analysis.
Line 120, "the results indicated,,," you can't use indicate here, change to another word.
Lien 121, "Abundance and biomass" where is the biomass? how do you calculate? Since the authors said abundance, but I can't find a data in the results, I advise to add some important data.
Line 123-124, "4 taxa" it's not clear, rewrite the sentence.
Line 148, change "," to "."
Line 151-152, "...up to 7 species....of the total species..." too complicated, and not clear, rewrite.
Line 183-184, "After the calculation,,, among four seasons.", delete.
Line 185-186, "The statistical ,,, not significant", delete.
Line 194, "significant difference." add F, p
Line 202, "structuring" change to another word, is not proper use.
Line 204-205, "...displayed a higher correlation with axes 2 than axes 1", how do you know? add the data please.
Line 230, "In past researches", delete, and change to in the last decades; "studied" changed to "reported"
Line 231, add the references after "...environmental factors".
Line 234-236, "Zhang et al. (2015, 2016),,,environmental factors" the sentence here is not closely connected with the text.
Line 245, "salinity" is the main factor, but in the research, Chla has the significant effects. As seen in Line 305, "... was not salinity but Chla...", Please discuss it further.
Line 318, the conclusion should be rewritten. It is similar to the abstract, but not. It should have some data in it.
Delete "In the summary"
Line 320, "quantity" ? please check
Line 323-326, "With the rapid development,,,, Yangtze estuary" Sine the research did not have data about the fishing intensity, you can't say. And the results said Chla is the main effecting factor, the data was related to industrialization and urbanization.
Table 4, many useless data. Would you please give a table about the summary statistics for the first two axes? such as percentage, cum percentage, cum constr. percentage for variation.,,,,

Annotated reviews are not available for download in order to protect the identity of reviewers who chose to remain anonymous.

Reviewer 3 ·

Basic reporting

1. Basic report: This paper reports the results of ichthyoplankton collections in the four seasons in the Yangtze Estuary and found that the ichthyoplankton assemblage was different among the seasons. An important finding of this paper by comparing the results with the previous studies in the Estuary, is the recent changes in ichthyoplankton abundance and composition in the study area, which seemed to be associated with the decline of freshwater runoff from Yangtze River.

Experimental design

2. Experimental design: Ichthyoplankton samplings were conducted in 4 seasons, in May, August, November, and February at the 40 sampling stations including river mouth and estuarine areas, which has sufficient coverage in terms of area and season. The trawl net was towed on the surface waters and therefore mostly collected pelagic eggs and larvae. If the net was towed obliquely from the near-bottom layer to the surface, the authors obtained different results in abundance and in species diversity. When we talk about fisheries resources, demersal fish species are also important. So in the next step, the author better attempt collections of demersal fish eggs and larvae.

Validity of the findings

3. Validity of the findings: The findings of this paper are limited to the year 2012. But by comparing the results of the previous papers, this paper showed an important change in ichthyoplankton abundance and composition in the Yangtze Estuary in recent years.

Additional comments

4. General comments: This paper is worth publishing in PeerJ after revising some points listed below.

The following points need to be reconsidered and revised if necessary.
1. “the Yangtze estuary” better be written “the Yangtze Estuary”, like Yangtze River.
2. Any difference between “ichthyoplankton community” and “ichthyoplankton assemblage”? In general, “community” has a structure like a kind of society but “assemblage” is only a simple gathering of animals. In this paper, description of abundance and composition of eggs and larvae collected by the trawl net seems to be in numbers and kinds of ichthyoplankton and nothing to do with the ecological relationship among the species. So the author better use “assemblage” rather than “community”.
3. “Research voyages” better be written “research cruises”.
4. No definite article to Latin names of animals.
5. Usually a definite article is not applied to season
6. Difference in ecotype is based on different habitats rather than ecological habit.
7. Line 121: biomass of Perciformes, need to show in table 1 which families belong to the order Perciformes.
8. Line 123: Should “4 taxa” be written as “4 ecotypes”?
9. Line 133: “for breeding and development” better be written as “for reproduction and development”.
10. Line 143: backish be corrected brackish. Similar in many of this word.
11. Line 150: E. japonicas be written as E. japonicus . Similar in many of this Latin name. Be careful with that the word processor automatically misspell this name from japonicus to japonicas.
12. Line 166: “in dominance degree of dominant species” be written as “the degree of dominance”.
13. Line 236: “species construction” may be written as “species composition”.
14. First paragraph of the Discussion: Are the authors mentioning that salinity in 2012 was higher than before due to decreased runoff of freshwater from the Yangtze River? If so, the author need to show salinity data in 2012 and in the previous years.
15. Line 280: “reproductive period” better be written as “reproductive season” or “spawning season”.
16. Line 287: “grown fish” is to be written as “adult fish”.
17. References: Abbreviations of author names and style of cited paper descriptions are to be uniform.
18. Table 1: Title of the table be “Presence (+)of species in ichthyoplankton samples in 2012. Scombrida be written as Scombridae. Hemirhamphus sajori be written as Hemiramphus sajori.
19. Table 3: footnote be written as “numbers with different superscript are significantly different each other”.
20. Figure 2: Sampling stations with no ichthyoplankton collection (0) need to be located by (x).

Reviewer 4 ·

Basic reporting

In this manuscript, the language is very clear. Comprehensive and sufficient references were cited in this manuscript and describe background of this study very well. The structure of this manuscript and content of tables and figures are fine, which meet the criterion of Journal. The authors supplied the raw data, which also reach the requirement of Journal. The results finding are well described.

Experimental design

The experimental design section met the requirement of Journal. However, the question for the research field is not clear. There is no recent research to discuss the ichthyoplankton community structure since 2012. The survey method was followed the Specification of Oceanographic Investigation. The details of data analysis were well described, which can be easily repeated.

Validity of the findings

Because of suitable survey method and appropriate data treatment, the result is robust and reliable. The conclusion is well stated and respond to the question cited in the introduction section.

Additional comments

Although advantage in the manuscript, there are some points needed to improve before the publication
1. In the introduction section, you cited the several references about the characteristics of ichthyoplankton community structure in the past. Please give major findings for the cited references especially references Liu et al, 2008; Wei et al, 2012.
2. In the discussion section, different years species construction of ichthyoplankton assemblages were compared, which showed a descending trend. The decline of ichthyoplankton species may relate the adult fish species variation. Please compare the adult fish assemblages of different years to support the variation in ichthyoplankton assemblages.

---

## Round 0.2 · Minor Revisions

In light of the experts' advice I am delighted to accept your manuscript. I invite you to revise your paper one last time to respond to the minor comments raised by one reviewer.

Reviewer 2 ·

Basic reporting

The revised Ms has been revised according to the suggestions. But the figs need a few changes.
Fig1, the map of China is lack of the South China Sea.
Fig5 is not professial, the labels and captal, please revise.

Experimental design

It's no problems.

Validity of the findings

It's no problems.

Additional comments

I accept the revised MS with minor changes.
Fig1, the map of China is lack of the South China Sea.
Fig5 is not professial, the labels and captal, please revise.

---

## Round 0.3 · Minor Revisions

Your article is scientifically Acceptable, however a final check of the manuscript by James Reimer, one of the Section Editors for this part of the journal, showed that it still needs another round of English editing. Please can we ask you to further edit the language, as PeerJ does not perform language editing as part of the production process.

---

## Round 0.4 · accepted · Accept

This is a valuable contribution to the understanding of ichthyoplankton communities in the Yangtze Estuary. Congratulations.

#